Published in FAST Workshop on Smalltalk Related Technologies (11/2022)

# Ordering Optimisations in Meta-Compilation of Primitive Methods

**Nahuel Palumbo**                                                    *nahuel.palumbo@inria.fr*
*Univ. Lille, Inria, CNRS, Centrale Lille, UMR 9189 CRIStAL, RMOD, F-59000 Lille, France*

**Guillermo Polito**                                                  *guillermo.polito@inria.fr*
*Univ. Lille, Inria, CNRS, Centrale Lille, UMR 9189 CRIStAL, RMOD, F-59000 Lille, France*

**Pablo Tesone**                                                      *pablo.tesone@inria.fr*
*Pharo Consortium*
*Univ. Lille, Inria, CNRS, Centrale Lille, UMR 9189 CRIStAL, RMOD, F-59000 Lille, France*

**Stéphane Ducasse**                                                  *stephane.ducasse@inria.fr*
*Univ. Lille, Inria, CNRS, Centrale Lille, UMR 9189 CRIStAL, RMOD, F-59000 Lille, France*

**Reviewed on OpenReview:** *https://openreview.net/forum?id=jYsMG5sjQy*

## Abstract

Meta-compilation schemes help to automatically build Just-in-Time (JIT) compilers from interpreters by performing a meta-interpretation of the VM interpreter. Generated JIT compilers face the well-known problem of phase ordering: selecting a good optimisation sequence to apply to the compiled programs. Manual optimisation lists are hard to maintain and are *one-size-fits-all* solutions that assume that a single sequence is equally effective in all possible programs. Generating such a list automatically is still challenging nowadays.

In this paper, we explore the phase-ordering problem in the case of the meta-compilation of Pharo VM interpreter primitives. In addition to a *manual* strategy, we present three automatic strategies to find good-enough optimisation sequences: a *search-based* approach, a *predictive* approach based on code shape, and an *automatically-found fixed list* approach. We compare them altogether by measuring the relative compiled code size and their rate of convergence. We evaluate this work over 17 of Pharo's language interpreter primitives. On average, the predictive strategy gives its optimal result before the rest with 21% fewer optimisations, the search strategy finds better results in complex cases. This article shows that automatic approaches seem promising for primitive meta-interpretation.

## 1 Introduction

Meta-compilation schemes help to automatically build Just-in-Time (JIT) compilers from interpreters (Section 2)(Rigo & Pedroni, 2006; Vergu & Visser, 2018). A meta-compiler performs a meta-interpretation of VM interpreter code to generate JIT compiler code. Such systems use meta-interpretation because they are implemented as an abstract interpreter interpreting a VM interpreter. The meta in meta-interpreter comes to the fact that they interpret an interpreter. The automatic generation of JIT compilers from an interpreter eases programming language implementation extensions: There is no need to be a compiler expert to be able to extend the JIT compiler. Moreover, we can build VMs for different languages by just providing different interpreters.

Generated JIT compilers face the well-known problem of phase ordering (Almagor et al., 2003; Ashouri et al., 2016): selecting a good optimisation sequence to apply to the compiled programs (Section 3). On the other

hand, manual optimisation lists are hard to maintain and are *one-size-fits-all* solutions that assume that a single sequence is equally effective in all possible programs. Generating such a list automatically is still challenging nowadays. The interrelation between optimisations is hard to predict (Almagor et al., 2003) and generates a large search space, raising the challenge of guiding the search and its stop condition (Almagor et al., 2004). Figure 1 illustrates the problem by showing the number of instructions in the meta-compiled *primitiveAdd* of Pharo's VM as optimisations are applied. The figure shows three sets of optimisations:

1. optimisations that reduce the number of instructions,

2. optimisations that in addition to (1), keep the same number of instructions, and

3. optimisations that in addition to (1) and (2), increase the number of instructions.

This example shows that different sequences of optimisations produce different results and that if the search stops at a local minimum it can prevent the obtention of better results. Existing work identifies the search space as discrete with many local minima but rare good solutions. Many use machine learning to identify optimisation sequences *e.g.,* genetic algorithms, predictive algorithms, biased random searches, or neural networks (Almagor et al., 2004; 2003; Kulkarni & Cavazos, 2012; Ashouri et al., 2016).

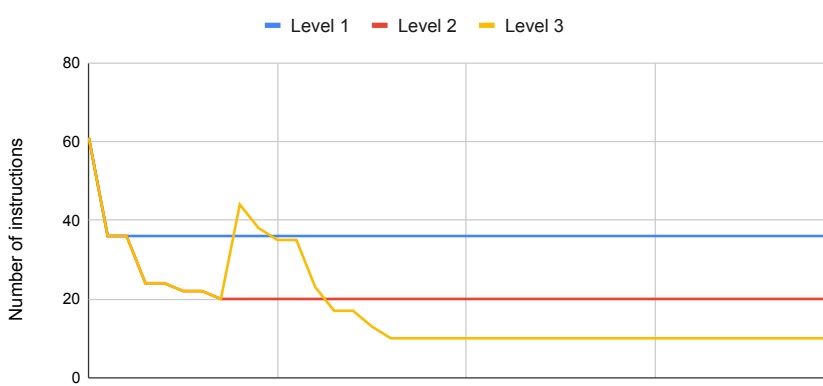

Figure 1: Trace of primitiveAdd with a search-based approach using the three heuristic levels. It shows the change in the number of instructions in the IR as selected optimisations are applied for each level of heuristic.

In this paper, we explore the phase-ordering problem in the case of the meta-compilation of Pharo VM interpreter primitives (Section 4). Until now, the meta-compilation scheme used a hand-written list that took months of iterated work by compiler developers. This list is hard to maintain and sometimes applies optimisations without effect in the compiled program. We use three alternative strategies to find good-enough optimisation sequences: a *search-based* approach, a *predictive* approach based on code shape, and an *automatically-found fixed list* approach. We compare them with the *manual* one by measuring the relative compiled code size and their rate of convergence.

We evaluate this work over 17 of Pharo's language interpreter primitives (Section 5). We show that all strategies arrive at the same optimal result for most primitives. On average, the *predictive* strategy gives its optimal result before the rest with 21% fewer optimisations. The *search* strategy finds better results in complex cases. None of them found an optimisation sequence better than the hand-written version in complex cases.

## 2 Context: Meta-compilation of primitives

The present work is implemented for Pharo (Ducasse et al., 2017), a dynamically-typed object-oriented programming language. In the Pharo Virtual Machine (VM), primitive definitions are methods written in Slang, a subset of Pharo itself, inside an Interpreter class (Miranda et al., 2018).

Our meta-compiler, called Druid, performs an ahead-of-time meta-compilation of VM interpreter primitives to generate a JIT compiler template for each of them. The meta-compilation works as a translation process that takes as input the language interpreter and produces JIT compiler code. Our meta-compilation approach uses meta-interpretation: it is implemented as an abstract interpreter interpreting the VM interpreter.

Internally, our meta-compiler uses an SSA-form Intermediate Representation (IR), a register-based control flow graph, where optimisations are applied. Optimisations perform mutations on the IR. We show a diagram of the architecture in Figure 2.

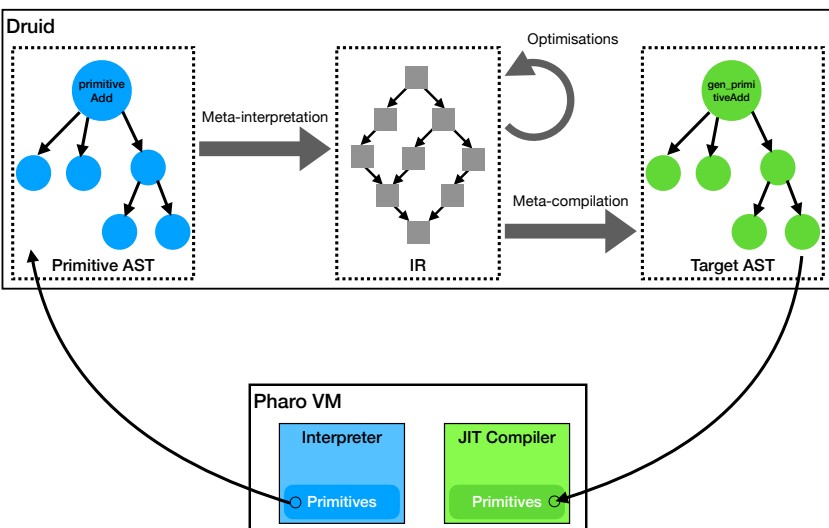

Figure 2: Architecture of the Druid meta-compiler. It receives as input the interpreter primitive definition AST. It builds an IR where optimisations are applied. Finally, it generates a new primitive definition for the JIT compiler.

In the end, the resulting IR is transformed into a JIT compiler template. This is a method inside a JIT compiler class. The template uses abstract registers and operations and, at run time the JIT compiler maps them to a specific target architecture and generates the expected machine code (Miranda, 2011).

Figure 3 illustrates the interpreter and JIT compiler code for the same *primitiveAdd*. This primitive performs the addition of integer objects with an overflow check:

- At the interpreter level, it will pop the last two values on the stack (argument and receiver), check that they are integer objects, calculate the addition of them, check it does not overflow, and finally push the result to the stack.

- The JIT compiler is activated when the system detects a *hot-spot*. It uses the template to generate machine code. The generated code performs the integer object checks on registers already loaded with the first argument and receiver values, calculate the addition checking overflow, and move the result to the expected register.

```
1   Interpreter >> primitiveAdd
2       <numberOfArguments: 1>
3
4       | maybeSmallInteger maybeSmallInteger2 result |
5
6       maybeSmallInteger := self stackValue: 0.
7       maybeSmallInteger2 := self stackValue: 1.
8
9       "Check small integer objects"
10      (objectMemory isIntegerObject: maybeSmallInteger)
11          ifFalse: [ ^ self primitiveFail ].
12      (objectMemory isIntegerObject: maybeSmallInteger2)
13          ifFalse: [ ^ self primitiveFail ].
14
15      "Check for overflow"
16      result := self
17          sumSmallInteger: maybeSmallInteger
18          withSmallInteger: maybeSmallInteger2
19          ifOverflow: [ ^ self primitiveFail ].
20
21      self pop: 2 thenPush: result
```

```
1   JITCompiler >> gen_primitiveAdd
2       | jump1 jump2 jump3 currentBlock |
3       "Check small integer objects"
4       self TstCq: 1 R: Arg0Reg.
5       jump1 := self JumpZero: 0.
6       self TstCq: 1 R: ReceiverResultReg.
7       jump2 := self JumpZero: 0.
8
9       self MoveR: Arg0Reg R: TempReg.
10      self SubCq: 1 R: TempReg.
11      self MoveR: ReceiverResultReg R: ClassReg.
12      self AddR: ClassReg R: TempReg.
13
14      "Check for overflow"
15      jump3 := self JumpOverflow: 0.
16      self MoveR: TempReg R: ReceiverResultReg.
17      self genPrimReturn.
18
19      "Fallthrough failling primitive"
20      jump1 jmpTarget: self Label.
21      jump2 jmpTarget: self Label.
22      jump3 jmpTarget: self Label.
```

Figure 3: Interpreter vs. JIT compiler primitive for integer addition. The interpreter definition works with values on the stack. The JIT compiler uses registers and machine code operations.

In both cases, if any check fails, the primitive fails and the virtual machine falls back to execute a normal message send routine.

## 3   Motivation for optimal optimisation sequences

To have an optimised JIT compiler template an optimising meta-compiler is needed. Finding an optimal optimisation sequence for a program is not trivial because the interrelation between them is hard to predict (Almagor et al., 2003). Some optimisations open new opportunities to others, so different orders usually arrive at different results.

Figure 4 shows an example of the compiler phase-ordering problem with three optimisations. One removes unused code (R), the second performs constant propagation without removing dead code (P) and the third duplicates basic blocks increasing code size but uncovering optimization opportunities (D). The final number of instructions in the IR of a primitive depends on the order that they are applied.

This dependency also implies that one optimisation could be applied multiple times in different moments. Solving this problem implies generating a large discrete search space with many local minima but few optimal solutions (Almagor et al., 2004). Guiding the search in this space and deciding when a solution is good enough is a hard and not intuitive task.

Our research questions are as follow:

- Is there one heuristic that is good enough to optimise our set of primitives?

- How many optimisations are necessary, at least, to arrive at the optimal version of each primitive?

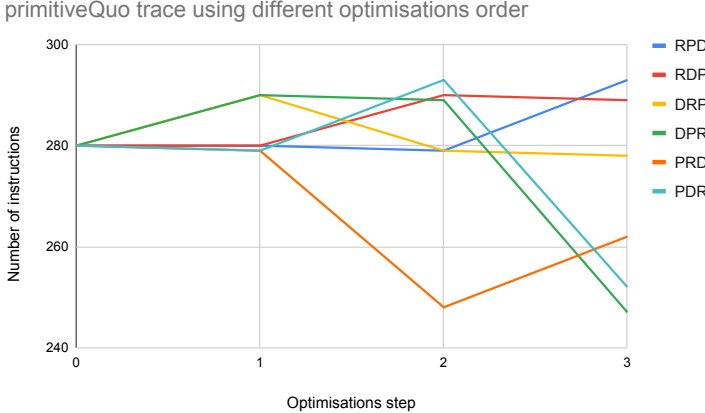

Figure 4: Optimisation order. Applying the same optimisations in different orders produces different results.

## 4 Comparing the four approaches

To answer our questions, we developed three different strategies to select the optimisation sequence for a primitive meta-compilation, that we compared to our pre-existent optimisation list hand-written by compiler developers (*manual* strategy). We analysed them by comparing the resulting IR version and optimisation sequences. The metric to compare IRs and to guide the search is the number of instructions of the Control Flow Graph. Thus, an IR is better than another if it can perform the same primitive computation with fewer instructions.

**Manual strategy.** This is a list of optimisations hand-written by experts based on their experience and knowledge about how optimisations work. This list took months of iterated work by compiler developers. Thus it is hard to maintain. This list is fixed, pre-calculated and the same for all primitives in a *one-size-fits-all* fashion. As this strategy is not guided nor profiled, there is no way to know when it arrives at its optimal result. This means that the optimisation list is always applied until the end, without effect in most cases. We will use this strategy as a baseline to compare evaluation results.

**Search strategy.** This is a heuristic-based search using a hill climber algorithm that selects an optimisation that reduces the number of instructions in the IR. It is an automatic approach, no need for optimisations experts to find a good optimisations sequence. This strategy is configured to find the first, last or best optimisation in a list. It computes all possible optimisations to the current IR and compares the resulting IR. Building search heuristics is hard, especially with complex IRs. It increases compilation time since it has to try many possible options in each stage.

| Search-based heuristic | | |
|---|---|---|
| **Level** | **Optimisation target** | **Post optimisations** |
| 1 | Reduce the number of instructions | - |
| 2 | Propagation and constants folding | DEAD CODE ELIMINATION |
| 3 | Code duplication | COPY PROPAGATION, SCCP, DEAD CODE ELIMINATION and CLEAN CONTROL FLOW |

Table 1: Different search-based heuristics. Each level describes the optimisation target and the post-optimisations to be selected by the hill climber algorithm.

The heuristic is based on a hill-climber algorithm with three levels of search presented in Table 1. In our approach, if there exist optimisations that do not improve the IR removing instructions but open new

opportunities to others, they are selected. In the end, if the algorithm does not find any optimisation at any level, then the search is finished and a Null optimisation is selected. A Null optimisation does not perform any change on the IR, it is not necessary to continue the search, thus it has arrived at its optimal result.

At level 1, it searches for optimisations that improve the IR for our metric. Only optimisations that reduce the number of instructions are selected here. If it does not find any optimisation, because no one can reduce instructions in the current state, the next configured levels are used.

At level 2, it searches for an optimisation that improves the IR for our metric after applying the optimisation and removing dead code. This allows the algorithm to select optimisations that do not remove instructions but left unused instructions, such as constant propagation and folding. When it does not find any optimisation, the next configured levels are used.

Level 3 takes into account optimisations that produce code duplication. As these optimisations increase the number of instructions, we consider the IR after copy propagation and folding and dead code elimination. This allows the selection of optimisations that duplicate code but open other opportunities.

Figure 1 shows the effect of each different level. Optimisations selected by level 1 always reduce the number of instructions. Optimisations selected by level 2 keep the same number of instructions but they decrease later. Optimisations selected by level 3 increase the number of instructions, where the graph move from 20 to 44 instructions (more than 100%), but at the end, the IR finishes with fewer instructions, in our example with 10 instructions (50% less). The stable value at the end represents the number of instructions of the program after applying all optimisations.

**Predictive strategy.** This strategy computes a list of possible optimisations based on IR form. It is an automatic approach that does not try every optimisation: the shape of the IR selects the corresponding optimisation to be applied. Table 2 shows the conditions for each optimisation that the IR should satisfy.

At first, this strategy evaluates the incoming IR and creates a list of possible optimisations. It applies all of them in any order. Once finished, it recomputes the list of possible optimisations using the current IR and repeats. This strategy ends when the list of possible optimisations to apply does not improve the current IR, or when the total number of optimisations arrives at a configurable limit.

| Predictive strategy conditions | |
| --- | --- |
| **Optimisation** | **IR Condition** |
| Branch Collapse | If there is a conditional jump without an inlined condition |
| Clean Control Flow | If there is a simple jump to a block with a unique predecessor |
| Copy Propagation | If there is a copy instruction |
| Dead Block Elimination | If there is a block without predecessors |
| Dead Branch Elimination | If there is a dead branch |
| Dead Code Elimination | If an instruction (with a result computation) has no users |
| Dead Edge Splitting | If there is a dead path |
| Failure Code Tail Duplication | If the exit primitive block has more than one predecessor |
| Phi Simplication | If there is a phi |
| Redundant Copy Elimination | If there is a copy between same physical register |
| SCCP | If an instruction lattice results in a constant value (a constant folding success) |

Table 2: Predictive strategy IR conditions for each optimisation.

**Automatically-found fixed list strategy.** This strategy uses a fixed list of optimisations automatically generated by the previous *search* strategy. We selected the largest generated optimisation list and evaluate it with all primitives. As it is based on a fixed list, this strategy applies all optimisations until the end, similarly to the *manual* strategy. This strategy does not need optimisation experts to find a good optimisation sequence, making it a *cheap* strategy that is calculated once and reused for many cases.

# 5   Evaluation

Currently, our meta-compiler supports the meta-compilation of the 17 primitives listed in Table 3. Table 4 shows all implemented optimisations with a small description and opportunities that it opens.

| Primitives | |
|---|---|
| **Name** | **Description** |
| primitiveAdd | Small integers addition with overflow check |
| primitiveSubtract | Small integers subtraction with overflow check |
| primitiveMultiply | Small integers multiplication with overflow check |
| primitiveLessThan | Small integers comparison |
| primitiveGreaterThan | Small integers comparison |
| primitiveLessOrEqual | Small integers comparison |
| primitiveGreaterOrEqual | Small integers comparison |
| primitiveEqual | Small integers comparison |
| primitiveNotEqual | Small integers comparison |
| primitiveDivide | Machine integers division |
| primitiveQuo | Machine integers quotient |
| primitiveBitXor | Bits xor |
| primitiveBitShift | Bits shift |
| primitiveFail | Failing primitive |
| primitiveMod | Small integers quotient with overflow check |
| primitiveDiv | Small integers division with overflow check |
| primitiveAt | Array access with bound check |

Table 3: Pharo VM interpreter primitives supported by our meta-compiler.

The search-based strategy we use is always the level 3 heuristic in three different versions: the first, the last and the best result that improves the metric. We applied all strategies to all supported primitives tracing the number of instructions after each optimisation, similar to Figure 1.

## 5.1   Optimal IR

For each strategy, we show the number of instructions after applying all optimisations. This number refers to the minimum number of instructions found by the strategy for each primitive. We see that all strategies arrive at the same number of instructions for all simple primitives. For complex primitives, small integer division and quotient and array access, all strategies arrive at an IR with more instructions than the *manual* approach. Figure 5 shows the number of instructions relative to the *manual* strategy for each primitive.

Out of the three variants of the search strategy, the Best configuration achieves better results, having on average 14% more instructions than the *manual* strategy. We will consider only this configuration in the rest of this paper.

Predictive and *automatically-found fixed list* strategies finish with 22% and 26% more instructions than *manual* strategy on average, respectively. It demonstrates that our heuristics are good enough for most supported primitives, they arrive at the same result, but they have problems taking decisions over complex scenarios.

## 5.2   Optimisation list

We measure how many optimisations were performed by each strategy to arrive at their optimal number of instructions in each trace, as illustrated in Figure 6. On average, Best *search* and *automatically-found fixed list* strategies arrive at each optimal IR by applying the same number of optimisations as the *manual* strategy. The *predictive* strategy arrives at its optimal IR with 21% fewer optimisations than the *manual* strategy. We identify that the *manual* strategy converges faster in the case of simple integer operations, this

| Optimisations | | |
|---|---|---|
| **Name** | **Description** | **Open opportunities** |
| BRANCH COLLAPSE | Inline conditions in conditional jumps | Dead path analysis |
| CLEAN CONTROL FLOW | Merge instructions in consecutive blocks to avoid unnecessary jumps | Better local analysis |
| COPY PROPAGATION | Replace copy instructions in operands by real value | Better dependency analysis and possible unused code |
| DEAD BLOCK ELIMINATION | Remove inaccessible blocks | - |
| DEAD BRANCH ELIMINATION | Remove branches with only dead paths | Better propagations |
| DEAD CODE ELIMINATION | Remove unused instructions | - |
| DEAD EDGE SPLITTING | Duplicate blocks with dead and not-dead paths | Create branches with only dead paths |
| FAILURE CODE TAIL DUPLICATION | Tail duplicate block with resulted code | Divide fail and success paths |
| PHI SIMPLICATION | Replace one operand phis with copy instruction | Better propagations |
| REDUNDANT COPY ELIMINATION | Remove copies of form x := x | - |
| SCCP | Performs constants folding and propagation | Better dependency analysis and possible unused code |

Table 4: Optimisations implemented in our meta-compiler. For each optimisation, we present a description and possible opportunities that it opens to other optimisations.

is probably because the optimisation list was created based on these primitives. In the cases of complex primitives, where not all strategies arrive at the same IR, we have different results.

It is important to note that this analysis is good to compare the rate of convergence of each strategy, but it does not answer which strategy will finish before. Remember that *manual* and *automatically-found fixed list* strategies must apply all optimisations until the end, while *search* and *predictive* strategies have to test many possible options in each state.

### 5.3 Results

We found that all strategies arrive at the same optimal IR in most primitives compilation. For those simple cases, Predictive and Best Search strategies achieve the optimal IR with 21% fewer optimisations than the *manual* strategy, on average. In complex cases, the *best search* strategy arrives at IRs with 14% more instructions than the *manual* strategy, on average. It is the closest strategy to the *manual* approach, taking a similar number of optimisations to arrive at the optimal IR.

The *best search* and *predictive* strategies calculate the next optimisation(s) based on the current IR state. These strategies add a searching time to the optimisation time. This is a trade-off between automatic optimisation selection and fixed optimisation list.

The *automatically-found fixed list* strategy is the cheapest option measured by avoiding search time without manual selection by the developers. For complex primitives, it arrives with 26% more instructions than the *manual* strategy, on average, but it keeps the same rate of convergence. The *manual* strategy arrived at better optimal results, but it is also the hardest to maintain.

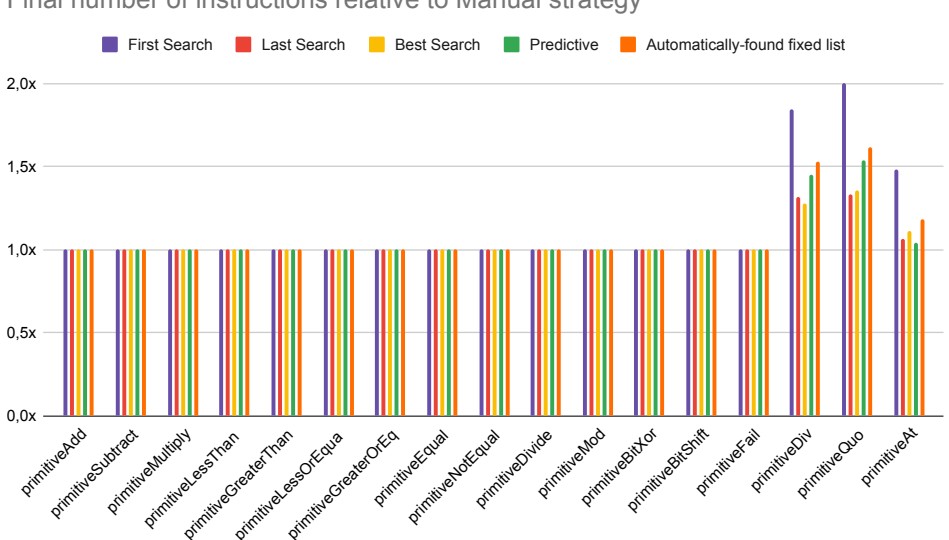

Figure 5: Number of instructions by strategy after applying all selected optimisations compare to *manual* strategy. Lower is better.

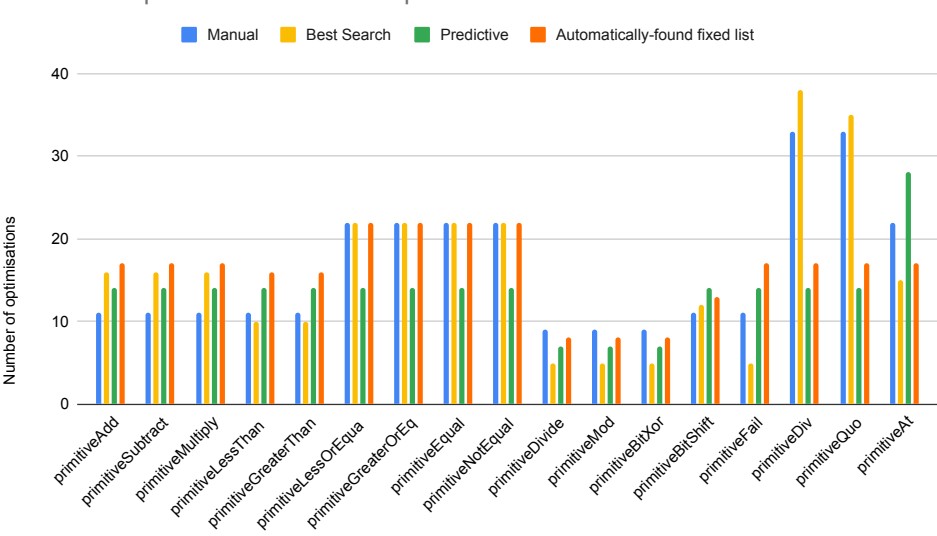

Figure 6: Number of optimisations to arrive at its optimal version. Lower is better.

## 6 Related work

Cooper et al. describe the problem of building a well-ordered list of optimisations in optimising compilers (Cooper et al., 2002). They identify the search space as discrete with many local minima but rare good solutions. They suggest solutions based on genetic algorithms, predictive algorithms, or biased random searches to improve a simple hill climber (Almagor et al., 2004). Kulkarni et al. explain how to improve the search time of optimisation sequences using genetic algorithms (Kulkarni et al., 2005).

Kulkarni and Cavazos describe some issues to solve the phase-ordering problem using genetic algorithms and propose a neuro-evolution technique to construct heuristics based on neural networks (Kulkarni & Cavazos, 2012). The survey (Ashouri et al., 2018) is a good description of state-of-the-art techniques to solve these issues using Machine Learning. Most of the work in this area is based on machine learning techniques.

Ashouri et al. propose a predictive trained model for speedup predictions based on a greedy Depth First Search heuristic (Ashouri et al., 2016). It can select the next-best optimisation improving the default LLVM-generated code by 2%.

Guo et al. developed an optimisation-specific search heuristic, based on specific knowledge about optimisations, and compares it with other generic searches (Guo et al., 2010). Their work is close to the one presented here.

Other research explores feedback-driven searches. Some use strategies to explore the optimisation space based on iterative compilation and many optimised versions of the same code (Triantafyllis et al., 2003). Others explore a profile-based approach based on an execution profiler (Chang et al., 1991).

## 7 Conclusion

In this paper, we compared four different strategies to select the list of optimisations to apply in a meta-compiler of primitive methods: *manual*, *best search*, *predictive* and *automatically-found fixed list*. We measured the number of instructions for each optimised IR and the number of optimisations necessary to arrive at it by each strategy.

We found that all strategies arrive at the same IR in most primitive compilations. On average, the automatic *predictive* strategy achieves its optimal IR with 21% fewer optimisations than the *manual* strategy built by compiler developers. In complex cases, on average, the *best search* strategy arrives at IRs with 14% more instructions than the *manual* strategy.

We have shown that good-enough compiled code for Pharo's primitives can be achieved by automatically selecting optimisations. Searching time is the trade-off between automatic searches and a fixed optimisation list maintained by compiler developers. But long-time analysis is not a big problem for this work, as it is an ahead-of-time task. We can search for a good solution without time constraints. We want to work in better heuristics for new search-based approaches.

As we are looking to increase the number of supported primitives and implemented optimisations, we are interested in an automatic approach to applying the optimisations. The next primitives will be more complex than the current ones, and we will need to implement new optimisations for them. With an automatic approach, VM developers will be free of maintaining the current manual list of optimisations for the meta-compiler, which is less trivial to understand in each development iteration.

Correlations between the number of instructions and the number of optimisations expose the presence of primitives with similar behaviour, thus similar IR. It suggests that a clustering-based approach (Martins et al., 2014) will allow reusing one optimisation sequence for many primitives. Maybe a mix of our *predictive* and *best search* strategies can be an option also.

### Acknowledgments

This work was funded by Inria's Action Exploratoire AlaMVic.

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
