# OpenReview forum: "Ordering Optimisations in Meta-Compilation of Primitive Methods"
_FAST.org.ar/2022/Workshop — FAST Smalltalk 2022_

### Official Review · Reviewer_Gzf2 · 2022-10-31
**Ordering Optimisations in Meta-Compilation of Primitive Methods**

**Rating:** 7
**Confidence:** 5

**Review:**

Many English as a second language errors. The most pervasive one is the word "archive" instead of "achieve".
abstract: "solve it" -> "solving it", "stop it" -> "stop", "archive" twice, "We make" -> "We do", "arrive to" -> "arrive at", "In average" -> "On average", "arrive to" -> "arrive at", "less optimizations" -> "fewer optimizations"
Several places where "big" should be "large"
"strategy divide the optimisations in three" -> "strategy divides the optimisations into three"
"that left the same" -> "that leave the same" and missing period at the end of that sentence
"and add" -> "and adds"
"bigger" -> "larger"
"over 17" -> "over 17 of"
"In average" -> "On average"
Generally describes the work that was done in present tense, where past tense would be more idiomatic. (But not for comments about the paper itself, which should be present tense.)
"arrive to different" -> "arrive at different"
"compare the resulted IR version" -> "compare the resulting IR version"
"less instructions" -> "fewer instructions" (fewer is for enumerable types, less is for e.g. water, flour, etc.)
"It is an heuristic" -> "This is a heuristic"
"arrive to" -> "arrive at"
"less instructions size" -> "fewer instructions"
"evaluates the incoming IR and create" -> "evaluates the incoming IR and creates"
"On finished" -> "Once finished"
"automatic generated" -> "automatically generated"
"Manual strategy. It is" -> "Manual strategy. This is"
"there is not way" -> "there is no way"
"so all optimisations list" -> "so the optimisations list" or "so all optimisations"
"In average" -> "On average"
I stopped doing this because it was distracting me from reviewing for content... but awkward expressions continue...
Figure 7 would be better as a number relative to manual... the +/- percentages were harder to interpret.
Actually, one of figures 7 or 8 is redundant... they show the same information, but 8 is more readable.
arggg... must correct...
"arrived to" -> "arrived at the"
"in the meaning of avoid searching" -> "measured by avoiding search"
"and to not decide the optimisations order by developers" -> "without the developers deciding a priori"
That whole paragraph is awkward.
"problem of build a" -> "problem of building a"
Second paragraph of Conclusions exhibits many, many of the problems above.

As the last paragraph of the Conclusion (indirectly) says, it's not clear that this is a problem that needs optimizing... it's ahead-of-time and there are only a couple hundred primitives, the various approaches are comparable, and all found the manual approach. Even if it took overnight to find a good solution for each primitive, using a dozen mschine you'd have all done in a week.

But it's a good paper (apart from the pedantic grammar problems identified above), it reviews the literature, and a negative result is still a result.

---

### Official Review · Reviewer_BvU6 · 2022-11-01

**Rating:** 7
**Confidence:** 3

**Review:**

This paper is about approaches to turning VM primitives, such as (for Smalltalk) defined in Slang, into efficient JIT templates. Its context is a meta-compilation approach where various standard compiler passes are provided over a register-based IR to which Slang can be compiled, and which can be lowered to an abstract syntax (template) representation of a JIT emission operation. The work presented considers how to replace a manually crafted list of these passes with some automatic search heuristics.

The main finding is that for most of the 17 Pharo primitives tried, a simple search can find a pass order yielding apparently efficient (at least in code size) JIT templates -- as good as the manual pass order. However, for a few it cannot (divide, quotient, array access).

The paper amounts to a worthwhile experience report about an interesting technique. I think this work will be of interest to some in the community. However, it's not motivated very well in the text. How much maintenance effort goes into the JIT templates for primitives? Is it really worth an elaborate meta-compilation system, including automatically ordered optimization passes, when a simpler and more manual approach might do? The answer might be 'yes' but the paper does not do much to explain this. Since the manually crafted pass order still seems to do best, this question remains in more than one sense.

I would have liked to see a clearer discussion of how other meta-level approaches, such as RPython's, perform analogous feats. It's possible the answer is "they don't", but even that is worth explaining.

==Smaller comments

I was not sure why the IR-generating step is 'meta-interpretation'. If it produces a register-machine IR, it seems to me like a kind of translation rather than interpretation.

When discussing the results, the word 'optimal' is misused. It should mean 'the best possible IR', but is used to mean 'the IR at which the heuristic terminates'. Perhaps I am missin something, but it seems the whole issue is that this is sometimes clearly not optimal.

The paper is in need of thorough proof-reading to correct spelling and grammar issues. I haven't itemised the problems, but two recurring issues are with conjugation of verbs ('the applied strategy divide', 'if no optimisation satisfy them') and possessives ('of Smalltalk VM primitives definition' => "of Smalltalk VM primitives' definitions' or "of the definitions of Smalltalk VM primitives") . On several occasions it says 'archive' for 'achieve'. There are many other issues.

In section 5, the selection of figures was really overkill, since they seem to be many ways of showing the same thing.

Table 5 is mostly redundant, but it does contain some useful elaboration of the motivation, in terms of code maintenance, effort by experts, etc. The conclusion also makes this motivating observation far better than the Introduction does; I recommend moving this discussion there. The current introduction is too 'in media res' and spends too long about a parallel with the analogous pass-ordering problem in general-purpose compilation pipelines. It's difficult (probably impossible) to transfer the present paper's findings into that more general context, so the analogy is overblown.